# COMPOSITIONAL 4D DYNAMIC SCENES UNDERSTANDING WITH PHYSICS PRIORS FOR VIDEO QUESTION ANSWERING

**Xingrui Wang**[1]**, Wufei Ma**[1]**, Angtian Wang**[1]**, Shuo Chen**[2]
**Adam Kortylewski**[3, 4]**, Alan Yuille**[1]
[1] Johns Hopkins University    [2] Tsinghua University
[3] Max Planck Institute for Informatics   [4] University of Freiburg

## ABSTRACT

For vision-language models (VLMs), understanding the dynamic properties of objects and their interactions in 3D scenes from videos is crucial for effective reasoning about high-level temporal and action semantics. Although humans are adept at understanding these properties by constructing 3D and temporal (4D) representations of the world, current video understanding models struggle to extract these dynamic semantics, arguably because these models use cross-frame reasoning without underlying knowledge of the 3D/4D scenes. In this work, we introduce **DynSuperCLEVR**, the first video question answering dataset that focuses on language understanding of the dynamic properties of 3D objects. We concentrate on three physical concepts — *velocity*, *acceleration*, and *collisions* within 4D scenes. We further generate three types of questions, including factual queries, future predictions, and counterfactual reasoning that involve different aspects of reasoning about these 4D dynamic properties. To further demonstrate the importance of explicit scene representations in answering these 4D dynamics questions, we propose **NS-4DPhysics**, a **N**eural-**S**ymbolic VideoQA model integrating **Physics** prior for **4D** dynamic properties with explicit scene representation of videos. Instead of answering the questions directly from the video text input, our method first estimates the 4D world states with a 3D generative model powered by physical priors, and then uses neural symbolic reasoning to answer the questions based on the 4D world states. Our evaluation on all three types of questions in DynSuperCLEVR shows that previous video question answering models and large multimodal models struggle with questions about 4D dynamics, while our NS-4DPhysics significantly outperforms previous state-of-the-art models. Our code will be available at https://github.com/XingruiWang/DynSuperCLEVR.

## 1 INTRODUCTION

Visual question answering (VQA) is a comprehensive task to assess how well machine learning models can identify objects, understand their relationships, and perform reasoning with multimodal input. When it comes to video question answering (VideoQA), models must not only capture the static features in frames but also understand temporal dynamics, such as object movements and interactions over time, especially in 3D space. Despite the recent progress in multimodal foundation models, understanding these dynamic features remains challenging (Ma et al., 2024).

A series of studies in cognitive science (Hamrick et al., 2016; Ullman et al., 2018) has found that humans excel at understanding the dynamics and interactions in the physical world. This enables humans to understand the spatial and temporal relationships between objects, predict future object states and interactions, and ultimately perform complex tasks such as planning and manipulation in the 3D world. For video question answering, it is also important to incorporate these dynamic features as an integral part of video understanding.

However, existing video question answering datasets have primarily focused on high-level temporal semantics, such as human activities (Fabian Caba Heilbron & Niebles, 2015; Goyal et al., 2017),

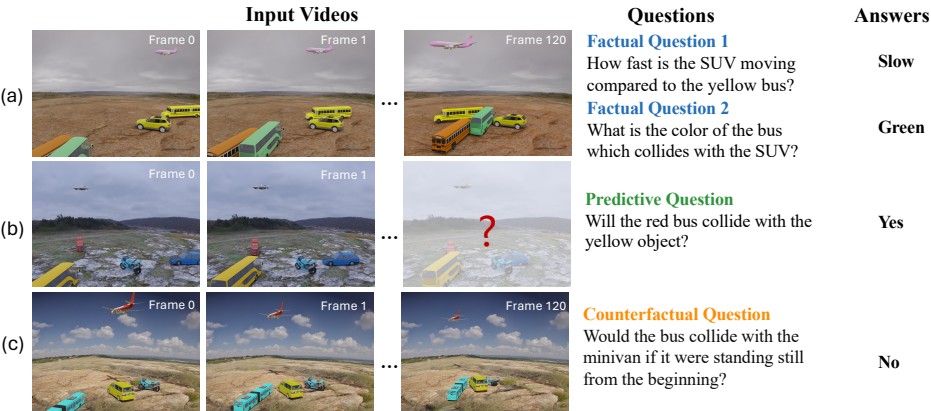

Figure 1: We propose DynSuperCLEVR to study the 4D dynamics properties of objects and their collisions. We also design three types of questions. Factual questions and counterfactual questions will take the whole 120 frames as input, while the predictive questions will take the first 30 frames.

due to a lack of 3D/4D annotations. Such evaluations are often susceptible to biases and shortcuts in natural videos, so models may perform quite well without truly understanding key dynamic properties (Ma et al., 2024). Another line of work exploits synthetic environments and studies how models understand dynamic and physical properties, given 3D ground truth obtained from simulators (Yi et al., 2019; Chen et al., 2022; Bear et al., 2021; Tung et al., 2023; Ates et al., 2022; Patel et al., 2022; Girdhar & Ramanan, 2020; Zheng et al., 2024). However, these datasets often consider only simplified settings, either lacking diverse physical properties in simulations (Ates et al., 2022; Yi et al., 2019) or lacking language integration (Bear et al., 2021; Tung et al., 2023). Moreover, these datasets employ rudimentary rendering with clear backgrounds and toy-like 3D assets, creating a significant domain gap between synthetic videos and real-world natural videos. While many conceptual findings have been derived from experiments on these datasets, these results lack practicality for understanding complex physical events in the real world. Without realistic rendering and natural language support, these datasets are also not suitable for analyzing the 4D dynamics understanding of recent large multimodal models.

To quantitatively study 4D dynamics understanding and to analyze the limitations of current video understanding models, we introduce **DynSuperCLEVR**, the first video question answering (VideoQA) dataset that focuses on the language understanding of the dynamic 4D properties of objects. Specifically, our dataset studies three crucial dynamic properties of 3D objects – velocities, accelerations, and collisions, and designs natural language questions for three types of reasoning, *i.e.*, factual, future prediction, and counterfactual. Notably, by introducing acceleration, we achieve a more complex physical scene compared to the previous data generator (Greff et al., 2022). To address the synthetic-to-real domain gap in previous video benchmarks (Yi et al., 2019; Tung et al., 2023) and enable zero-shot evaluation of large multimodal models on our dataset, we improve the realism of our videos by introducing diverse appearances of objects and backgrounds with new textures.

To demonstrate the importance of explicit scene representation in answering these 4D dynamics questions, we present **NS-4DPhysics**, a neural-symbolic model that reasons about these dynamics by first estimating explicit 4D scene representations. Our NS-4DPhysics consists of two key modules: a 4D scene parsing module followed by a symbolic reasoning module. The 4D scene parsing module learns a robust 3D generative model (Ma et al., 2022; Wang et al., 2023b) of the scene and captures the dynamics of objects with a physical prior. With a 3D generative representation of the scene, our scene parsing module is robust to partial occlusion and generalizes well to unseen 3D poses and object appearances during training, which are very common in natural videos.

We test a wide range of state-of-the-art VideoQA models on our DynSuperCLEVR dataset, including neural symbolic models (Yi et al., 2019; Wang et al., 2024), standard end-to-end models (Perez et al., 2018), large multimodal models (Xu et al., 2024; Lin et al., 2023), and the proprietary model GPT-4o. Results show that these models struggle to answer challenging questions about dynamic properties of the 4D scenes due to a lack of explicit knowledge of the 3D world or the 3D dynamics of objects. Meanwhile, our NS-4DPhysics outperforms previous state-of-the-art models by a wide margin across

different dynamic properties and types of reasoning questions. This demonstrates the importance of developing an explicit 4D dynamics representation for multimodal agents to understand physical events presented in the videos and to foresee upcoming events.

Our contributions are as follows: (1) We present DynSuperCLEVR, the first VideoQA dataset that focuses on language understanding of the dynamic properties of objects (velocity, acceleration) and multi-object interactions (collisions). (2) We propose NS-4DPhysics, a neural-symbolic model that first reconstructs the 4D scene with a dynamic 3D generative model with physical priors as the perception module and then reasons about dynamics over the explicit 4D scene representation. (3) Extensive results on different settings of DynSuperCLEVR reveal key pitfalls of state-of-the-art VideoQA models, including those powered by LMMs and industrial training data. We further demonstrate that NS-4DPhysics outperforms other VideoQA baselines by a wide margin, showing the advantages of learning explicit 4D dynamic representations.

## 2 RELATED WORK

**Video question answering.** Video question answering (VideoQA) is a challenging task because models must not only detect and identify objects from static images but also track and infer objects' changes and interactions over a sequence of frames. A number of VideoQA datasets annotate question-answer pairs on natural videos (Xue et al., 2017; Yang et al., 2022; Majumdar et al., 2024; Wu et al., 2024). However, these datasets are not suitable for studying dynamic physical properties, as natural videos contain limited object interactions. Another line of work focuses on physical reasoning in simulated environments (Yi et al., 2019; Chen et al., 2022; Ding et al., 2021; Tung et al., 2023). However, these datasets are either built in restricted settings (Yi et al., 2019; Chen et al., 2022) or lack real dynamics and forces common in the real world (Tung et al., 2023). In order to further challenge and explore the limitations of current video-text models on dynamics reasoning, we develop DynSuperCLEVR with improved realism for both video quality and physics simulation.

**Video-text models.** With the availability of web-scale image-text or video-text paired datasets (Schuhmann et al., 2022; Bain et al., 2021), recent video-text models (Wang et al., 2022; Lin et al., 2023; Xu et al., 2024) have adopted heavy multi-modal pretraining and achieved improved results on a wide range of video understanding tasks. However, these models often exploit biases and shortcuts for dynamic reasoning, failing to capture the physical intrinsics that drive object movements and interactions. Our NS-4DPhysics incorporates a 4D dynamic scene representation in our scene parser, allowing compositional reasoning of various physical events in the video sequence.

**Physical scene understanding.** Understanding the physical events and inferring the future state requires a model to apply physical principles to explain the observed events or simulate the future outcomes. Previous studies explored physics engines for simulation and learning (Kadambi et al., 2023; Battaglia et al., 2013), or integrating a differentiable physics engine in deep learning models (Wu et al., 2015; 2017; Kyriazis & Argyros, 2013). Other works learn physical properties from a compositional scene (Hamrick et al., 2016; Ullman et al., 2018; Ding et al., 2021; Chen et al., 2022; Zheng et al., 2024). However, previous methods were limited to simple scenes and we extend our scope to more realistic scenes with real-world objects and complex simulations, enabling physical scene reasoning with our dynamic 3D compositional reasoning.

**3D generative models.** Our model is built on top of previous 3D generative models for image classification (Jesslen et al., 2023), 6D pose estimation (Ma et al., 2022), and 3D-aware visual question answering (Wang et al., 2023a). In image understanding, these generative models learn a compositional representation of 3D objects with feature activations on vertices, conducting the scene understanding tasks with analysis-by-synthesis. As found in Wang et al. (2023a), the advantage of these methods is the robustness in recognizing 3D objects from scenes, compared with the discriminative baseline such as Faster R-CNN (Ren et al., 2016). We extend the previous 3D compositional models to the video understanding task, which learns a 4D dynamic scene representation from video. Such representation allows us to reconstruct world states at each timestep, analyze the trajectories of objects, and reason about the physical events.

## 3 DATASET

To investigate 4D dynamic properties in VideoQA tasks, we introduce the **DynSuperCLEVR** dataset. This synthetic dataset provides fully annotated 4D scene structures for videos, along with the reasoning steps to answer corresponding questions. We concentrate on the 4D dynamic properties of objects — `velocity` and `acceleration` in 3D scenes and their comparison. We also introduce

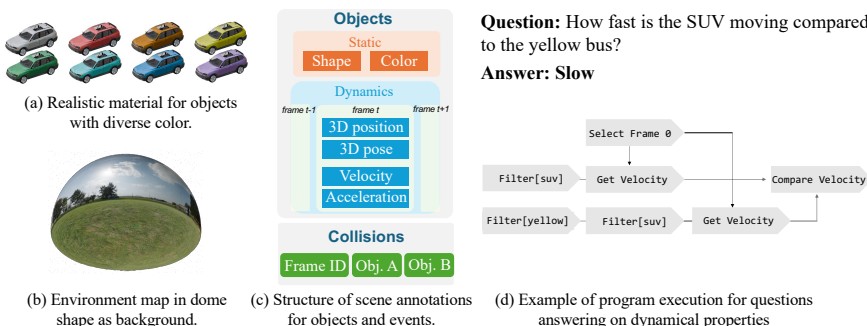

(a) Realistic material for objects with diverse color.

(b) Environment map in dome shape as background.

(c) Structure of scene annotations for objects and events.

(d) Example of program execution for questions answering on dynamical properties

Figure 2: An illustration of the construction of DynSuperCLEVR. (a) We use the same 3D object meshes from SuperCLEVR but generate more realistic textures for different colors. (b) The background is created by mapping a real image environment map onto a dome shape. (c) Our video data is fully annotated with 4D dynamic scene structure, containing static and dynamic properties for objects and collision components. (d) For each question, we design new operation programs for 4D dynamic properties, which can be executed on the scene structure to answer the questions.

the `collision` event in 3D space, which is more challenging than in the 2D ground plane. The collision demonstrates how the dynamic properties will affect the interactions among objects.

As an advanced video reasoning task, we design questions across three levels — (1) **factual questions**, which query properties and events directly from the visible frames, (2) **predictive questions**, which require forecasting future collisions based on the current 4D dynamics and scenes, and (3) **counterfactual questions**, which challenge models to imagine how changes in the 4D dynamics would alter the outcomes.

In this section, we first introduce the three key components for designing the video data in our proposed dataset: (1) the design of more realistic textures for objects and backgrounds (Sec.3.1), (2) the 4D dynamical scene annotations involving 4D dynamic properties for objects (Sec.3.2) and their collision events, and (3) the rendering of the video and physical simulation process (Sec.3.3). Then, in Sec.3.4, we describe our three types of questions at different reasoning levels and the reasoning program for question answering.

## 3.1 3D SCENE CONSTRUCTION

**3D Objects.** WWe use the same 3D objects from SuperCLEVR (Li et al., 2023), which include five vehicle categories (car, plane, bicycle, motorbike, bus) with 21 sub-types, but we regenerate the textures for each mesh model to improve realism. For each color label (gray, red, brown, yellow, green, cyan, blue, and purple), we create 6 variants and change the face colors of the object meshes. Based on the part annotation in SuperCLEVR, only the primary body receives color edits, while others are updated with natural elements like black wheels, transparent windows, or red tail lights on cars (see Fig. 2a). More examples of all textured 3D objects are provided in the Appendix A.

**Backgrounds.** Compared with the previous synthetic dataset (Yi et al., 2019; Li et al., 2023), we change the background to highly realistic images and lights. As shown in Fig. 2 (b), we use real HDRI images[1], which contain real captured images with environment maps in 509 different scenes. We map the image as material onto a dome shape as the background to create a 3D view.

## 3.2 4D DYNAMICS SCENE ANNOTATIONS

After defining the shape and color of objects as static attributes, we assign them dynamic properties, allowing them to move and interact within the 3D physical world. Each object is defined by its 3D position, pose, velocity, and acceleration, as described below (see Fig. 2(c)).

**3D Position and pose:** The 3D position $(x, y, z)$ records the exact coordinates of the object at each timestep, specifying its location within the scene. The pose $(\alpha, \beta, \gamma)$ describes the object's orientation and rotation in 3D space, defining how the object is aligned relative to its surroundings.

---

[1]https://polyhaven.com/hdris

**Velocity.** The `velocity` $(v_x, v_y, v_z)$ represents the change in the object's 3D position over time, capturing both the speed and direction of motion. As one of the properties included in the question answering, we define three velocity states: `static` (0 m/s), `slow` ($\leq$ 3 m/s), and `fast` ($\geq$ 3 m/s) for language description. Given that the exact velocity is often difficult to determine from video alone, we ensure that each scene includes at least one object moving within each velocity state to provide clear distinctions.

**Acceleration.** The `acceleration` $(a_x, a_y, a_z)$ represents the rate of change of velocity over time in 3D space. Similar to the label for velocity, we define two types of motion for acceleration. Along the x-axis, objects can either accelerate due to an internal engine force or decelerate due to friction. In the z direction, the objects can either maintain a constant height (zero acceleration) or experience downward acceleration due to gravity. In our dataset, we simplify it into a binary label: `accelerating` or not, and `floating` or not.

**Collision Events.** Due to perceptual illusions, reasoning about collisions in 3D space is more challenging than in a 2D plane. Each collision involves two `objects`, and we record the `frame ID` when the collision occurs. The collision between objects is a direct result of their 4D dynamic properties. In our dataset, we capture not only the collision events in a given frame but also future collisions from simulation and counterfactual scenarios, where changes in the 4D dynamic properties at the beginning lead to a new sequence of collision events. This forms the foundation for the three question types, which will be introduced in Sec. 3.4.

### 3.3 PHYSICAL SIMULATION AND VIDEO GENERATION

The video data is rendered using a combination of the physics engine PyBullet and the Blender renderer. Building on the generation pipeline of Kubric (Greff et al., 2022), we integrate support for the new acceleration feature. At each timestep, we capture the 3D position, pose, velocity, and acceleration of all objects, along with all collisions between objects, as described above.

Each scene has a corresponding counterfactual version. We re-simulate the scene, randomly selecting one object's 4D dynamic property (velocity or acceleration) to modify its initial value. We then record the new property values and the resulting sequence of collision events caused by the changes.

### 3.4 QUESTIONS GENERATION FOR 4D DYNAMICS PROPERTIES

Following previous video question answering datasets (Yi et al., 2019; Chen et al., 2022), we develop question templates and generate three types of questions for factual, predictive, and counterfactual reasoning in our DynSuperCLEVR (see Fig. 1). Each question is paired with a corresponding program execution for reasoning (see Fig. 2(d)). Below, we describe the design of the three question types, while the full list of programs and more examples of program execution can be found in the Appendix C.

**Factual Questions.** Factual questions focus on the direct understanding of static and dynamic properties of objects, their comparisons, and collision events within 3D space in given frames. For our newly proposed 4D dynamic properties, we define questions that query the velocity and acceleration states of objects or compare their velocities in 3D space at specific moments. These moments are either defined by the start of a frame or the point when a collision event occurs. The questions about collisions involve predicting whether two objects have collided or identifying which objects are involved in a collision. We introduce new programs *query_velocity*, *compare_velocity*, and *query_moving_direction* for velocity; *query_accelerating* and *query_floating* for acceleration.

**Predictive Questions.** Predictive questions require forecasting future collision events that are not present in the video. In our DynSuperCLEVR, it is crucial for the model to first understand the positions, velocities, and accelerations of objects in 3D space from the video, and then apply reasoning to predict their future trajectories.

**Counterfactual Questions.** Counterfactual questions challenge the model to reason about hypothetical scenarios by changing the initial 4D dynamic properties of objects. This requires the model to have a strong understanding of how changes in dynamic features (velocity or acceleration) can affect the world states and cause collisions, and to apply these to re-simulate the potential outcomes contrary to the video input.

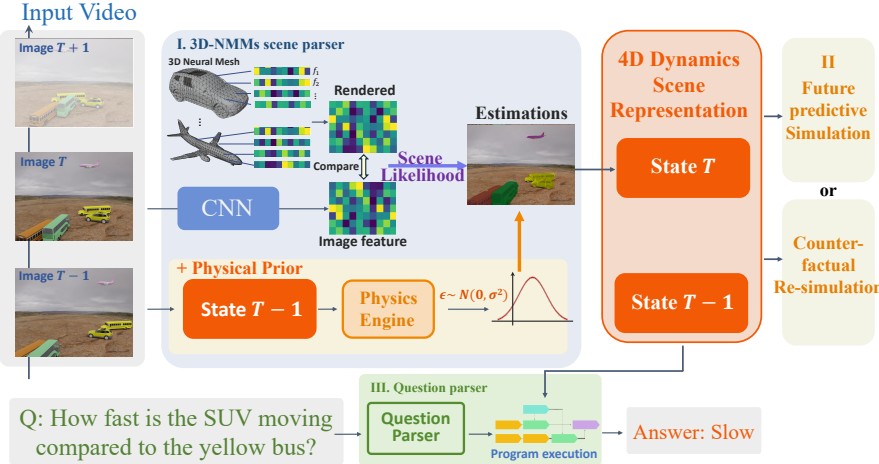

Figure 3: Our NS-4DPhysics has three main components. I: A **3D neural mesh scene parser**, which combines the rendering likelihood with a physics prior to parse the video into a 4D dynamic scene representation. II: The future states or counterfactual states can be simulated with the reconstruction result by the physics engine. III: A **question parser** that processes questions into reasoning programs and then executes the program over the predicted scene representation to answer the questions.

### 3.5 DATASET STATISTICS

We generate 1,000 video clips for training, 100 for validation, and 100 for testing. Each video is 2,000 ms long with a frame rate of 60, resulting in 120 frames per scene. In total, we create 7,850 factual questions, 2,750 predictive questions, and 989 counterfactual questions. Factual questions are open-ended and can be answered with a few words, while predictive and counterfactual questions are structured as true or false, requiring the model to determine if events will occur.

## 4 MODEL

In this section, we introduce NS-4DPhysics, a neural symbolic VideoQA model for solving questions about object dynamics in 3D space. As illustrated in Fig. 3, the model combines a **3D-aware scene parser** with a **probabilistic physical prior module** to convert video sequences into an explicit 4D scene representation. This explicit representation enables interpretable reasoning, allowing the model to predict future or counterfactual world states through physics simulation. For question answering, we employ a **question parser** to first convert the questions into executable programs, which are then used to answer the questions step by step based on the predicted 4D scene representation.

In the following sections, we introduce the 4D symbolic scene representation in Section 4.1, the scene parser with the physical prior in Section 4.2, future and counterfactual simulation in Section 4.3, and the language model and reasoning program in Section 4.4.

### 4.1 4D SYMBOLIC SCENE REPRESENTATION

The 4D dynamics scene representation is the world states used to reconstruct the video scene annotation shown in Fig. 2c. For each object $O$, we define its static attributes as **shape** $s^i$ and **color** $c^i$. The dynamic properties at each timestep $t$ consist of **3D position** $T_t$, **rotation** (pose) $R_t$, **velocity** $v_t$, and **acceleration** $a_t$, where each is a 3D vector. For an input video, each image frame at timestep $t$ is denoted as $\mathcal{I}_t$. The 4D symbolic scene representation $\mathcal{S}$ is defined as a sequence of object states $\mathcal{S}_t = \{O_t^1, O_t^2, \ldots, O_t^N\}$ over time, where $N$ is the number of objects in the scene.

### 4.2 DYNAMIC SCENE PARSER WITH PHYSICS PRIOR

The scene parser takes the videos as input and estimates the 4D dynamic world state $\mathcal{S}_t$ at time step $t$. This involves predicting the 6D poses of objects in frames, predicting their temporal properties (e.g., velocity and acceleration), and reasoning about collision events. In this work, we adopt a 3D generative model that learns a feature representation of the scene and predicts 3D world states from an analysis-by-synthesis perspective. We extend the work of Ma et al. (2022); Jesslen et al. (2023) to natural videos by employing a sequence decoding approach and adding a physical prior in our 4D scene representation for the model to understand real-world physical events.

**3D-NMMs Scene Parser**   3D neural mesh models (3D-NMMs) (Ma et al., 2022; Jesslen et al., 2023) learn a generative model to produce the 3D feature representation of the objects and parse the scene with analysis-by-synthesis. For each object shape, we train an object mesh $M_s = \{v_i \in \mathbb{R}^3\}_{i=1}^N$ with a neural texture $T_s = \{f_i \in \mathbb{R}^c\}_{i=1}^N$, where $s$ is the object category (shape), $N$ is the number of vertices, and $c$ is the feature dimension. A 3D-NMM can generate the 3D-aware scene representation by rendering the neural mesh model $O_s = (M_s, T_s)$ in the given 6D pose $\alpha$: $F_s(\alpha) = \mathfrak{R}(O_s, \alpha) \in \mathbb{R}^{H \times W \times c}$, with soft rasterization (Liu et al., 2019). By comparing the rasterization of 3D features and the 2D image feature, we can learn the neural textures with the ground truth category $s$ and 6D poses $\alpha$, or estimate $\alpha$ and $s$ during inference. We formulate this render-and-compare process as an optimization of the likelihood model:

$$p(F \mid O_s, \alpha_s, B) = \prod_{i \in \mathcal{FG}} p(f_i \mid O_s, \alpha_s) \prod_{i \in \mathcal{BG}} p(f_i' \mid B) \tag{1}$$

where $\mathcal{FG}$ and $\mathcal{BG}$ are the set of foreground and background locations on the 2D feature map and $f_i$ is the feature vector of $F$ at location $i$. Here, the foreground and background likelihoods are modeled as Gaussian distributions.

In our NS-4DPhysics, $\alpha$ is the 3D position $T_t$ and rotation $R_t$ of each object for time step $t$, which is estimated independently for each video frame (i.e., at the image level). The training phase learns the CNN extractor for feature $F$, the neural mesh $O_s$, and the background feature $B$ jointly (see Appendix D.1 for details). During inference, by maximizing the likelihood function in Eq. 6 on each frame, we can obtain the $T_t$ and $R_t$ for the predicted objects.

**Physical Prior**   The dynamical 3D world states predicted from the image frame do not consider the consistency in the time dimension. As our algorithm predicts $\mathcal{S}_t$ for each time step sequentially by maximizing the likelihood function in Eq. 6, one strategy is to initialize $\mathcal{S}_t$ with the predicted $\mathcal{S}_{t-1}$ in gradient descent. However, this only fits the case when the objects are moving uninterruptedly. In complex dynamical 3D scenes, the trajectory of objects can change dramatically due to their interactions, such as collisions. These require comprehensive prior knowledge of the physical world.

It's nontrivial to directly model the physical functions. Many previous studies incorporate physical engines within neural networks, but only apply to simple shapes like cubes, as the complex shapes of 3D objects and the interaction process are hard to model. On the other hand, computational physical engines (Coumans & Bai, 2016) excel at modeling physical functions for any known shapes but are not differentiable and are challenging to integrate into the de-rendering process during inference.

In our method, we modify a discriminative physics engine $\text{PE}(\cdot)$ into a probabilistic model by assuming $(R_t, T_t) = \text{PE}(\hat{R}_{t-1}, \hat{T}_{t-1}) + \varepsilon$ after obtaining $\hat{R}_{t-1}$ and $\hat{T}_{t-1}$. Here, we take $\varepsilon \sim \mathcal{N}(0, \sigma^2 I)$. Inspired by the idea from Bayes-Kalman filtering (Salzmann & Urtasun, 2011), we integrate a physics prior of $(R_t, T_t)$ into the likelihood function as a correction term after Eq. 6. Formally, the prior $q(R_t, T_t \mid \hat{R}_{t-1}, \hat{T}_{t-1})$ can be computed as:

$$4 \tag{2}$$

where $\mu = \text{PE}(\hat{R}_{t-1}, \hat{T}_{t-1})$, and $C = 1/\sqrt{(2\pi)^k |\sigma^2 I|}$.

Finally, the rotation $R_t$ and translation $T_t$ can be estimated by maximizing the joint likelihood of the rendering likelihood in Eq. 6 and the physical prior likelihood in Eq. 2:

$$\hat{R}_t, \hat{T}_t = \arg \max_{R_t, T_t} p(F \mid O_y, T_t, R_t, B) \cdot q(R_t, T_t \mid \hat{R}_{t-1}, \hat{T}_{t-1}). \tag{3}$$

**Dynamics Attributes.** After estimating the world states $(R_t, T_t)$, we compute the dynamics attributes `Velocity` and `Acceleration` by calculating the differences between consecutive translations and velocities over time. To reduce noise, a moving average filter with a window size of 5 is applied.

**Static Attributes.** `Shape` is determined by selecting the mesh model category with the highest likelihood (Wang et al., 2024). For `Color` prediction, we crop the object's region from the RGB image and train an additional CNN classifier for color recognition.

**Collisions.** Collisions are obtained from the physics engine $\text{PE}(\cdot)$ at the time of estimating the physical prior. For each collision, we record the time and objects involved.

### 4.3 FUTURE AND COUNTERFACTUAL SIMULATION

From the explicit 4D scene representation, we can easily simulate the future or counterfactual states of the objects. For either case, we apply the physics engine and assign the predicted translation, rotation, velocity, or counterfactual condition to all objects as initial conditions and simulate. We visualize the re-simulation results in Sec. 5.4.

### 4.4 QUESTIONS PARSING AND PROGRAM EXECUTION

After obtaining the 4D scene representation from the scene parser, we can parse the question into reasoning programs, which are then executed on the scene representation to predict the answer. The question parser follows previous work (Yi et al., 2019; Chen et al., 2022), where an LSTM sequence-to-sequence model is trained to parse the question into its corresponding program.

## 5 EXPERIMENTS

### 5.1 EXPERIMENT SETUP

**Baseline Models.** We select the representation models from the following three categories as baseline models on DynSuperCLEVR. (1) Simple classification-based methods: CNN+LSTM and FiLM. We extract frame-level features and average them over the time dimension. We encode questions with the last hidden state from an LSTM and then concatenate the two features to predict answers. (2) Neural symbolic models: NS-DR and PO3D-VQA, which represent the model with explicit 2D/3D scene representation for question answering, compared with our 4D representation. (3) Video large language models (Video-LLMs): including a fine-tuned InternVideo, and Video-LLaVA, PLLaVA, GPT-4o for zero-shot evaluation with in-context learning. Please refer to the appendix for detailed implementations.

**NS-4DPhysics Implementation.** The dynamic scene parser and CNN classifier are trained on the images, using 120k images for training. The dynamic scene parser uses ResNeXT as the feature extractor and is trained for 50 epochs with a batch size of 4 on 4 GPUs. The attributes classifier uses ResNet50 and is trained with the cropped images from the ground truth bounding box. During inference, we set the variance of the physical prior to 3.

### 5.2 VIDEO QUESTION ANSWERING RESULTS

We first compare the NS-4DPhysics with baseline models for the VideoQA task of DynSuperCLEVR. The results are shown in Tab. 1. For GPT-4o, we also prompt GPT-4o to first describe the video, then reason step-by-step before answering questions (GPT-4o+reasoning).

Table 1: Performance on the DynSuperCLEVR testing split for each question type: factual, predictive, and counterfactual. Factual questions are further divided into sub-types: **Vel**ocity, **Acc**eleration, and **Col**lision, with "**All**" representing overall accuracy. The average is taken as the overall accuracy across the three question types. [†] indicates GPT-assisted zero-shot evaluation. The highest performance among all baseline models is indicated by underlined values.

| | Average | Factual | | | | Predictive | Counterfactual |
| | | All | Vel. | Acc. | Col. | | |
|---|---|---|---|---|---|---|---|
| CNN+LSTM | 48.03 | 40.63 | 41.71 | 56.79 | 25.37 | 56.04 | 47.42 |
| FiLM (Perez et al., 2018) | 50.18 | 44.07 | 48.58 | 53.09 | 26.87 | 54.94 | 51.54 |
| NS-DR (Yi et al., 2019) | 51.44 | 51.44 | 55.63 | 46.34 | 46.86 | - | - |
| PO3D-VQA (Wang et al., 2024) | 62.93 | 61.22 | 62.21 | 73.17 | 51.20 | 65.33 | 62.24 |
| InternVideo (Wang et al., 2022) | 52.62 | 51.07 | 59.29 | 49.08 | 36.06 | 54.74 | 59.18 |
| Video-LLaVA [†] (Lin et al., 2023) | 38.09 | 37.04 | 37.62 | 52.76 | 23.56 | 38.78 | 40.88 |
| PLLaVA [†] (Xu et al., 2024) | 59.24 | 54.61 | 55.00 | 63.80 | 46.63 | 67.52 | 73.47 |
| GPT-4o[†] | 51.59 | 50.82 | 51.19 | 57.67 | 44.71 | 54.38 | 50.00 |
| GPT-4o + reasoning [†] | 56.06 | 55.50 | 58.81 | 57.67 | 47.12 | 56.93 | 58.16 |
| **NS-4DPhysics** | **82.64** | **87.70** | **88.66** | **83.73** | **88.46** | **85.71** | **74.51** |

**Comparison with Classification-Based Methods.** CNN+LSTM only reaches 40.63%, 56.04%, and 47.42% for factual, predictive, and counterfactual questions, while FiLM achieves 44.07%, 54.94%, and 51.54%. The performance of these models is notably lower than that of NS-4DPhysics, which achieves 87.70%, 85.71%, and 74.51% for factual questions, respectively. As classification-based methods generally rely on extracting features through CNNs, we demonstrate significant advancements in handling dynamic content with explicit scene representation.

**Comparison with the Neural-Symbolic Methods.**   The comparison between the Neural-Symbolic models shows significant performance improvement as the dimension of scene representation increases. PO3D-VQA constructs explicit 3D scene representation, leading to better estimation of the objects' positions and poses, which is beneficial for inferring the dynamics and interactions compared with the 2D presentation in NS-DR (9.78%) for factual questions. By pushing the dimension into 4D and considering the physical prior knowledge, our NS-4DPhysics achieves 82.64% overall accuracy, which is 19.71% higher than PO3D-VQA.

**Comparison with Video-LLMs.**   Although Video-LLMs demonstrate strong video understanding and generalization abilities, they underperform on DynSuperCLEVR. In zero-shot settings, Video-LLaVA achieves an overall accuracy of 38.09%, while PLLaVA performs better at 59.24%, excelling in predictive and counterfactual reasoning among the baselines. Even with fine-tuning, InternVideo reaches only 52.62%, lagging behind our NS-4DPhysics. These results indicate that existing video foundation models struggle to reason effectively in complex dynamic scenes.

## 5.3   ANALYSIS

We further analyze the effectiveness of two key components of NS-4DPhysics: the physics prior and the symbolic reasoning model. (1) Without the physics prior, the 4D world states are estimated using only the rendering likelihood, with time consistency maintained by initializing $\mathcal{S}_t$ from the previous frame, $\hat{\mathcal{S}}_{t-1}$. The result in Table 2 shows that adding the physics prior to the 4D scene representation significantly improves VideoQA performance. We also visualize the predicted world states of both models in Sec. 5.4 with the failure cases. (2) We also compare a variant of GPT-4o, where, as a language model, it processes the predicted scene structure from NS-4DPhysics as text input instead of raw images. However, the LLM still underperforms in 4D dynamic reasoning, highlighting the advantages of our symbolic reasoning model in capturing dynamic object properties.

Table 2: Analysis of the physics prior module and symbolic reasoning (SR) in our model. We compare our variants without the physics prior and augment LLM with explicit 4D scene representation.

|  | Average | Factual | | | | Predictive | Counterfactual |
|---|---|---|---|---|---|---|---|
|  |  | **All** | Vel. | Acc. | Col. |  |  |
| 4D Representation + SR (Ours) | **82.64** | **87.70** | **88.66** | **83.73** | **88.46** | **85.71** | **74.51** |
| w/o Physics Prior + SR | 75.97 | 79.68 | 81.40 | 81.30 | 74.88 | 78.83 | 69.39 |
| 4D Representation + GPT-4o | 61.39 | 65.49 | 66.67 | 54.60 | 71.63 | 57.14 | 51.09 |
| Video + GPT-4o | 56.06 | 55.50 | 58.81 | 57.67 | 47.12 | 56.93 | 58.16 |

## 5.4   QUALITATIVE RESULTS

We visualize how our model achieves more accurate results in reconstructing scenes and simulating future scenes for predictive questions, assisted by the physics prior. Fig. 4 shows an example of a factual question. Without the physics prior (row c), the model fails to track the correct position of the minivan and provides the wrong answer regarding the collision. Fig. 5 illustrates a predictive

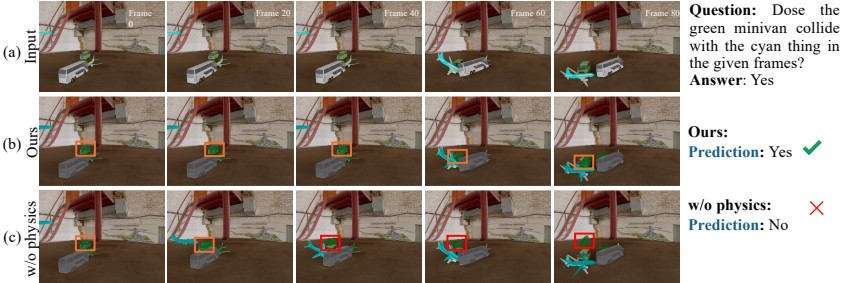

Figure 4: Qualitative examples of factual questions. (a) shows the input video; (b) Our NS-4DPhysics provides a better estimation for the motion with the physical prior. (c) The error in position predicted by the baseline w/o physics in the red box leads to a mistake in the answer.

question. Our model predicts more accurate positions for all the objects (b1), and the simulation of future frames in (b2) shows no collisions. However, without the physics prior, the error in 3D position prediction (especially for the school bus) causes all the vehicles to collide in future frames.

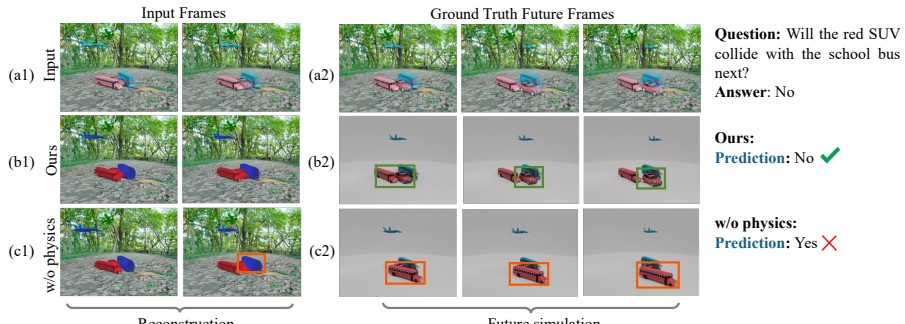

Figure 5: Qualitative examples of predictive questions. (a1) The first 30 frames are given to models as input video, and (a2) the following frames are hidden as ground truth future states; (b1) our NS-4DPhysics has a better estimation of the poses of objects, and (b2) provides a plausible simulation by re-simulation. (c1) The red box shows the error in pose estimation of the bus when lacking a physics prior, which (c2) causes the red SUV to collide with the school bus in the future.

**Analysis of Video-LLM.** To analyze the reasoning failures in Video-LLM, we examine two failure cases of PLLaVA and GPT-4o. (a) Misunderstanding the location and collisions in 3D space: Both PLLaVA and GPT-4o incorrectly predict that the truck collides with the fighter jet. In reality, the truck only collides with the purple articulated bus, while the plane is near the truck but does not make contact. This error results from relying on 2D projection, which fails to capture the correct 3D spatial relationships. (b) Failure to predict future 4D dynamics world states: Both models identify the objects correctly but fail to understand the objects' motion from the given frames. Although no collision occurs in the current frames, the bus is moving toward the airplane and will collide in the future—something the models are unable to predict.

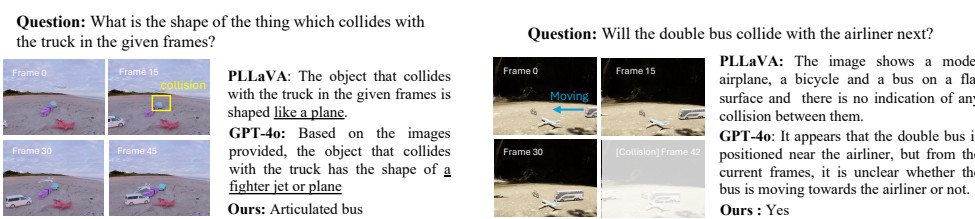

(a) Misunderstand the location and collisions in 3D space     (b) Fail to predict future 4D dynamics world states

Figure 6: Two failure cases of PLLaVA and GPT-4o. (a) Both models incorrectly predict a truck colliding with a fighter jet due to misunderstanding 3D spatial relationships, likely caused by reliance on 2D projection; (b) While the models correctly identify objects, they fail to predict future collisions due to a lack of understanding of the 4D dynamics of objects from the given frames.

## 6    CONCLUSION AND DISCUSSION

In this work, we explore 3D dynamic properties in video question answering. We introduce DynSuperCLEVR to study the 4D dynamics of scenes in VideoQA tasks with improved realistic textures. Our findings show that existing video-language models, even those with large-scale pretraining, struggle to capture critical 4D dynamic properties necessary for temporal and predictive reasoning. To address this, we develop NS-4DPhysics, a neural-symbolic model that estimates an explicit 4D scene representation and answers questions through program execution. Experimental results on DynSuperCLEVR demonstrate that our approach significantly outperforms previous state-of-the-art methods, highlighting its effectiveness in inferring dynamic properties and predicting future states. In ongoing work, as Kaushik et al. (2024) has demonstrated the potential of 3D NMMs for domain adaptation, we are studying how our NS-4DPhysics can be extended to work with real images. **Limitations.** As a synthetic VideoQA dataset, our DynSuperCLEVR currently focuses only on rigid objects with linear velocity and acceleration. More complex objects, such as articulated and non-rigid objects with complex trajectories, are not yet considered.

ACKNOWLEDGEMENT

This work is supported by ONR with award N00014-23-1-2641, from ARL Army Research Laboratory with award W911NF2320008 2812. Adam Kortylewski acknowledges support via his Emmy Noether Research Group funded by the German Reserach Foundation (DFG) under Grant No. 468670075. We appreciate the anonymous reviewers for their valuable feedback. We also thank Zhuowan Li and Qing Liu for their insightful discussions during the early stages of this project.

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
