# A  3D OBJECTS AND COLORS

In DynSuperCLEVR, we have 21 object classes and 8 colors. As described in the main paper, we improve upon previous datasets by generating new 3D textures with different colors. Below, we present all the 3D objects, with one object selected in each color.

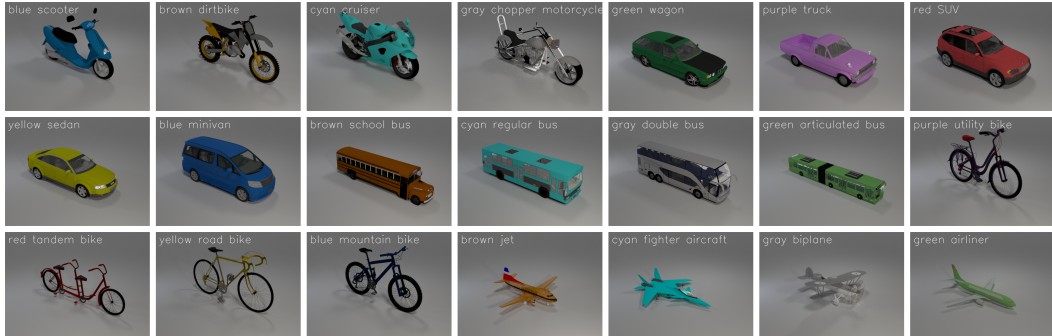

Figure 7: Examples of 3D objects in DynSuperCLEVR with a selected color for each object.

# B  DYNAMIC PROPERTIES SETTING

When constructing DynSuperCLEVR, we need to set the extra physical properties for objects for physics simulation. In the main paper, we introduced the design of the 4D dynamic properties, such as position, velocity, and acceleration. Here, we describe how the exact values are set in the physics engine in Table 3. Additionally, we need to set a series of physical properties in the physics engine, such as mass, friction, and restitution.

(1) **Mass and Gravity:** The mass of each object is calculated based on its specific shape model, which is linearly related to its volume with a density of 2.7. All objects in the dataset are subject to gravity, which influences their vertical movements and impacts when in flight or during falls. The gravitational constant is set to 10.

(2) **Friction:** Frictional forces affect the movement of all objects, especially when they interact with the ground or each other, slowing them down and eventually bringing them to a stop. The friction of objects is set to 0.2, while the friction of the floor (the dome-shaped background) is set to 0.4.

(3) **Restitution:** When objects collide or drop to the ground, elastic forces come into play, defined by elasticity coefficients. These forces affect how objects bounce off each other or rebound from barriers, significantly altering their trajectories and speeds. The restitution of both the objects and the floor is set to 0.5.

Table 3: Dynamic settings of DynSuperCLEVR

| Attribute | Description | Aeroplanes | Others |
|---|---|---|---|
| Mass | Calculated from volume | $\rho V$ | |
| Position | $(x, y)$ | Beta distribution | |
| | $z$ | Uniform distribution | 0 |
| Orientation | Faces to the center with noises | - | |
| Velocity | Initial state: static, slow, fast | $\{0, 3, 6\}$ m/s | |
| Internal Force | Engine force (Forwards) | $\{1, 0\}$m/s$^2$ | |
| | Floating force(Upwards) | $\{10, 0\}$ Mass $\times$ m/s$^2$ | 0 Mass $\times$ m/s$^2$ |

## C  PROGRAM EXECUTION

In our DynSuperCLEVR, we introduce new programs for the 4D dynamic properties and the collision events. In Table 4, we list all the programs and their input/output types involved in the DynSuperCLEVR.

Table 4: All operations and their input / output types involved in the DynSuperCLEVR

| Type | Operation | Input Type | Output Type |
|---|---|---|---|
| Event Operations | `filter_collision` | CollisionEventSet, Object | CollisionEventSet |
| | `get_all_col_partners` | CollisionEventSet, Object | ObjectSet |
| | `get_frame` | CollisionEvent | FrameID |
| | `come_in_frame` | Object | FrameID |
| Object filter Operations | `filter_attributes` | ObjectSet | ObjectSet |
| | `filter_static` | ObjectSet, FrameID | ObjectSet |
| | `filter_moving_velocity` | ObjectSet, FrameID | ObjectSet |
| | `filter_accelerating` | ObjectSet, FrameID | ObjectSet |
| | `filter_floating` | ObjectSet, FrameID | ObjectSet |
| Object Query Operations | `query_attributes` | Object | Shape or color |
| | `is_static` | Object, FrameID | Bool |
| | `query_moving_velocity` | Object, FrameID | Velocity |
| | `query_moving_direction` | Object, FrameID | Direction |
| | `is_accelerating` | Object, FrameID | Bool |
| | `is_floating` | Object, FrameID | Bool |
| Object comparison | `faster_velocity` | Object, Object, FrameID | Bool |
| | `slower_velocity` | Object, Object, FrameID | Bool |
| Input Operations | `Objects` | None | ObjectSet |
| | `Events` | None | CollisionEventSet |
| | `futureEvents` | None | CollisionEventSet |
| Counterfactual Operation | `Counterfactual_static` | Object | CollisionEventSet |
| | `Counterfactual_moving_slow` | Object | CollisionEventSet |
| | `Counterfactual_moving_fast` | Object | CollisionEventSet |
| | `Counterfactual_accelerating` | Object | CollisionEventSet |
| | `Counterfactual_floating` | Object | CollisionEventSet |
| Others | `unique` | ObjectSet | Object |
| | `exist` | ObjectSet | Bool |

Based on the programs above, we generate the corresponding program execution for each factual, predictive, and counterfactual question. Here, we provide more examples of the program execution in Fig. 8.

## D  DETAILS OF 3D NNMs SCENE PARSER WITH PHYSICS PRIOR

The ultimate goal of the scene parser is to estimate all dynamic states $\mathcal{S}_t$ from the corresponding observation $\mathcal{I}_t$ and the previous states $\mathcal{S}_{<t}$ using a probabilistic model. Previous work in 6D pose estimation has proposed using render-and-compare to maximize scene likelihood for static images. For the video data, we further consider the temporal consistency and physical plausibility of the dynamics estimation.

$$\hat{\mathcal{S}}_t = \arg\max_{\mathcal{S}_t} p\left(f(\mathcal{I}_t)|\mathcal{S}_t\right) \cdot p(\mathcal{S}_t|\mathcal{S}_{<t}), \tag{4}$$

$$\hat{\mathcal{S}}_0 = \arg\max_{\mathcal{S}_0} p(f(\mathcal{I}_0)|\mathcal{S}_0). \tag{5}$$

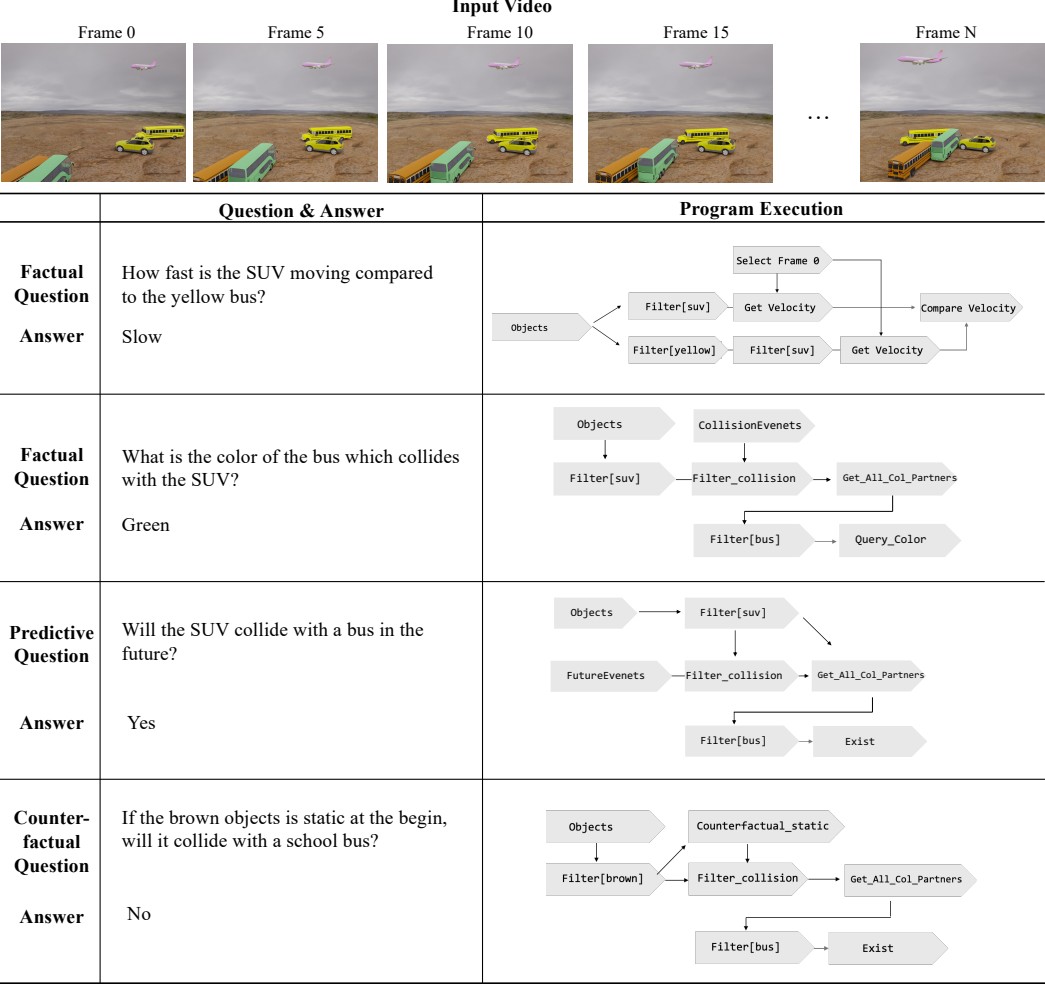

Figure 8: Examples of the reasoning program. From the input questions, we provide new operation programs to answer these questions step by step using a 4D dynamic scene representation defined in the main paper.

## D.1 3D NEURAL MESH MODEL

Inspired by the 3D-neural-meshed-based generative model for pose estimation methods on static images Wang et al. (2021); Ma et al. (2022), we develop a de-rendering-based generative model to reconstruct the 4D dynamic scene representation frame by frame. To ensure the predictions are not only plausible in 3D positions but also adhere to physical rules, we incorporate a physical likelihood model and a 3D generative model with rendering likelihood, as shown in Fig. 3 in the main paper. Here, we describe more details of the 3D neural mesh models as a preliminary.

**Preliminaries.** In previous work for static images Ma et al. (2022), Neural Mesh models were introduced for 6D pose estimation through inverse rendering. For that task, the goal is to jointly estimate the 6D pose (2D location $(x, y)$, distance $d$ to the camera, and 3D pose $(\alpha, \beta, \gamma)$) of objects in an image by comparing the Neural Mesh features after rendering with the input image features and maximizing the rendering likelihood. More formally, the mesh for a given object in category $c$ is represented as $M_c = \{v_i \in \mathbb{R}^3 | i = 1 \dots N\}$, where $v_i$ represents the vertex. The corresponding neural texture of the mesh $M_c$ is $T_c \in \mathbb{R}^{N \times l}$, where $l$ is the dimension of the feature. Thus, the neural mesh model for category $c$ is their aggregation $O_c = \{M_c, T_c\}$. The render-and-compare process is

formulated as an optimization of the likelihood model:

$$p(F \mid O_c, \alpha_s, B) = \prod_{i \in \mathcal{FG}} p(f_i \mid O_c, \alpha_c) \prod_{i \in \mathcal{BG}} p(f_i' \mid B) \tag{6}$$

where $\mathcal{FG}$ and $\mathcal{BG}$ are the sets of foreground and background locations on the 2D feature map, and $f_i$ is the feature vector of $F$ at location $i$, produced by the CNN feature extractor $\Phi$. Here, the foreground and background likelihoods are modeled as Gaussian distributions.

In this work, we transform the position into a world coordinate instead. Given the object 3D rotation $R = (\alpha, \beta, \gamma)$ and translation $T = (x, y, z)$, we can render the neural mesh model $O_c$ into a feature map $F_c$ with soft rasterization Liu et al. (2019).

**Training:** During training, we follow (Ma et al., 2022) and first jointly train a CNN feature extractor $\Phi$, the neural texture $\{T_y\}$, and the background model $B$. We utilize the EM-type learning strategy originally introduced for keypoint detection in CoKe (Bai et al., 2023). Specifically, the feature extractor that produces $f_i$ is trained using stochastic gradient descent, while the parameters of the generative model $\{T_y\}$ and $B$ are trained using momentum updates after every gradient step in the feature extractor, which has been found to stabilize training convergence. The final loss function is defined as a constructive loss between the features of the vertices:

$$\mathcal{L}_{\text{contrastive}} = -\sum_{i \in \mathcal{FG}} \sum_{j \in \mathcal{FG} \setminus \{i\}} \|f_i - f_j\|^2 - \sum_{i \in \mathcal{FG}} \sum_{j \in \mathcal{BG}} \|f_i - f_j\|^2 \tag{7}$$

**Inference as Scene Parsing** For static images, previous work Wang et al. (2021) has shown that the rendering likelihood can be used to estimate the 6D pose of objects. In our dynamic scene, similar strategies can be applied to each frame at time step $t$. As the neural mesh models are probabilistic generative models of neural feature activation, we can first define the rendering likelihood of the feature map $F_t$ given any 6D pose $R_t, T_t$ as:

$$p(\mathcal{I}_t | \mathcal{S}_t) = p(F_t \mid O_c, R_t, T_t, B) \tag{8}$$

$$= \prod_{i \in \mathcal{FG}} p(f_t^{(i)} \mid O_c, R_t, T_t) \prod_{j \in \mathcal{BG}} p(f_t^{(j)} \mid B), \tag{9}$$

where $\mathcal{FG}$ and $\mathcal{BG}$ are the sets of foreground and background locations on the 2D feature map, and $f_t^{(i)}$ is the feature vector of $F$ at location $i$ at timestep $t$. $B$ is a background parameter learned from training. Here, the foreground and background likelihoods are modeled as Gaussian distributions.

### D.2 Physical Prior

Rendering likelihood alone is insufficient to reconstruct a physically plausible 4D dynamic scene representation. We also integrate a physics prior into our likelihood model. The physical process can be modeled as a Markov model Salzmann & Urtasun (2011), where the physical prior distribution of rotation $R_t$ and translation $T_t$ at time step $t$ can be expressed as:

$$p(\mathcal{S}_t | \mathcal{S}_{<t}) = q(R_t, T_t \mid \hat{R}_{t-1}, \hat{T}_{t-1}). \tag{10}$$

However, it is nontrivial to directly model the physical functions. Most current studies on physical engines within neural networks focus only on simple shapes like cubes, as the complexity in shapes of objects and the interaction process is hard to model. On the other hand, computational physical engines such as Bullet excel at modeling physical functions for any known shapes but are not differentiable and are challenging to integrate into the de-rendering process during inference.

In our method, we modify the discriminative physics engine into a probabilistic model by introducing an uncertainty term $\varepsilon \sim \mathcal{N}(0, \sigma^2)$. Denoting a physics engine as $\text{PE}(\cdot)$, we can assume:

$$(R_t, T_t) = \text{PE}(\hat{R}_{t-1}, \hat{T}_{t-1}) + \varepsilon, \tag{11}$$

$$(R_t, T_t) \mid (\hat{R}_{t-1}, \hat{T}_{t-1}) \sim \mathcal{N}(\text{PE}(\hat{R}_{t-1}, \hat{T}_{t-1}), \sigma^2 I), \tag{12}$$

$$q(R_t, T_t \mid \hat{R}_{t-1}, \hat{T}_{t-1}) = C \exp\left(-\frac{1}{2\sigma^2} [(R_t, T_t) - \mu]^T [(R_t, T_t) - \mu]\right), \tag{13}$$

where $\mu = \text{PE}(\hat{R}_{t-1}, \hat{T}_{t-1})$ and $C = 1/\sqrt{(2\pi)^k |\sigma^2 I|}$.

Finally, the rotation $R_t$ and translation $T_t$ can be estimated by maximizing the joint likelihood of the rendering likelihood and the physical prior likelihood:

$$\hat{R}_t, \hat{T}_t = \arg\max_{R_t, T_t} p(F_t \mid O_c, R_t, T_t, B) \cdot q(R_t, T_t \mid \hat{R}_{t-1}, \hat{T}_{t-1}). \tag{14}$$

**Relationship to Bayes-Kalman Filtering** The conceptual ideas are directly inspired by the classic ideas of Bayes-Kalman filtering, where the goal is to estimate a hidden state based on a sequence of observations. Bayes-Kalman filtering consists of updating a probability distribution of the hidden state by a prediction step followed by a correction step that incorporates evidence from a new observation. It uses a dynamic model for how the hidden state changes with time, which is directly analogous to our physical prior. It has an observation model, corresponding to our scene parser, for how new observations provide evidence for the hidden state. Bayes-Kalman filtering, however, is difficult to implement in complex applications like ours because it requires us to represent and update a complex probability distribution. The standard approach for doing this is particle filtering, where the probability distribution is represented by a set of point particles that are updated during the prediction and correction steps. This is also challenging; thus, instead, we use a simple approximation that essentially uses a single particle. In future work, we will experiment to see if our model gives even better results if we use particle filtering instead of this approximation.

## E  DOMAIN EXTENSION FOR THE NS-4DPHYSICS TO THE REAL VIDEO

We design a case study to demonstrate the real-world generalization ability of the proposed NS-4DPhysics pipeline. Following the architecture of the proposed model, we train the 3D scene parser on the Pascal3D+ dataset, which contains 3D pose annotations but lacks object appearance labels. The qualitative reconstruction results, as shown in Fig. 9(a), demonstrate accurate estimations on the real video, where 4D dynamic properties, including velocities and accelerations, can be effectively inferred. Although the model is not trained on object appearances, its capabilities can be extended by incorporating object classifiers with proper annotations or by enabling open-vocabulary recognition through pretrained large vision-language feature embeddings (e.g., CLIP). This as an important direction in our future work.

Additionally, as shown in Fig. 9(b) and (c), similar types of questions can be posed for the given video, which can then be answered by executing the program step-by-step.

## F  DETAILS OF THE BASELINE MODELS

We select 7 representative models from the following 3 categories as baseline models on DynSuperCLEVR.

**(1) Simple Classification-Based Methods** encode videos with a CNN backbone and predict answers with a classifier head. Specifically, we consider **CNN+LSTM**, which aggregates frame-level CNN features and encodes the question using an LSTM, and **FiLM** Perez et al. (2018), which incorporates a feature-level linear modulation module for question answering.

**(2) Neural Symbolic Models** first parse the scene into object instances and then execute a program for question answering. **NS-DR** Yi et al. (2019) adopts Mask R-CNN for object detection. We modify it with the new program in DynSuperCLEVR and evaluate it on factual questions. Since their 2D simulator is unable to reason about objects in 3D, we do not compare it on predictive and counterfactual questions. **PO3D-VQA** Wang et al. (2024) uses a 3D detector and reconstructs an explicit 3D scene representation for each frame. We extend the model for VideoQA by computing dynamic properties from object locations and predicting collisions by filtering the distance between objects.

**(3) Large Multimodal Models** leverage large-scale image-text or video-text data for pretraining and achieve strong generalization abilities across various video-text tasks. We consider **Video-LLaVA** Lin et al. (2023) and **PLLaVA** Xu et al. (2024) for zero-shot evaluation, where a GPT model is used to evaluate the correctness of free-form answers. Additionally, we fine-tune a pretrained **InternVideo**

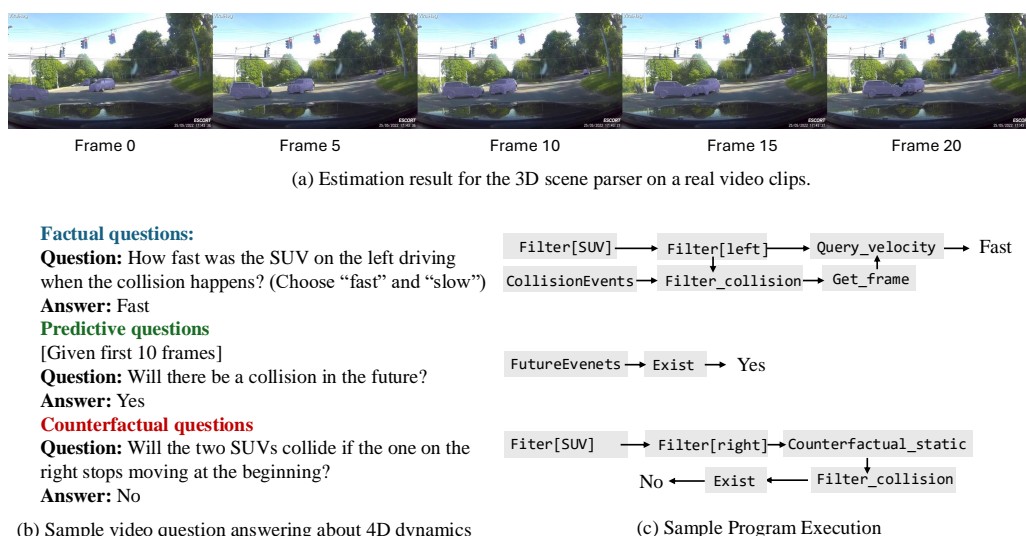

(a) Estimation result for the 3D scene parser on a real video clips.

**Factual questions:**
**Question:** How fast was the SUV on the left driving when the collision happens? (Choose "fast" and "slow")
**Answer:** Fast
**Predictive questions**
[Given first 10 frames]
**Question:** Will there be a collision in the future?
**Answer:** Yes
**Counterfactual questions**
**Question:** Will the two SUVs collide if the one on the right stops moving at the beginning?
**Answer:** No

(b) Sample video question answering about 4D dynamics

```
Filter[SUV] ─────→ Filter[left] ────→ Query_velocity ──→ Fast
                                            ↑
CollisionEvents ─→ Filter_collision ─→ Get_frame

FutureEvenets ──→ Exist ──→ Yes

Fiter[SUV]    ───→ Filter[right]─→Counterfactual_static
                                          ↓
          No ←── Exist ←── Filter_collision
```

(c) Sample Program Execution

Figure 9: Qualitative reconstruction results from real video data. (a) Estimation results from the 3D scene parser, where 4D dynamic properties, including velocities and accelerations, are effectively inferred. (b) Example questions about the video, which can be answered by executing the corresponding program step-by-step as shown in (c).

Wang et al. (2022) model that predicts answers using a classifier head. Finally, we evaluate the proprietary model **GPT-4o** by accessing it through the OpenAI API with customized system prompts. Specifically, we consider two settings: (i) asking the GPT-4o model to treat the problem as a multiple-choice question and predict the answer directly (see Figure 10); and (ii) asking the GPT-4o model to think step by step, provide necessary reasoning about the dynamic and physical events, and then give the answer to the question (see Figure 11).

**Zero-shot evaluation of GPT-4o with multiple choice questions**

**System:**
You are an intelligent chatbot designed for answering questions based on eight frames obtained from a video.

Your task is to analyze the frames of the video, identify the nature of the object movements (such as speed, acceleration, and direction), and then determine the answer to the question.

INSTRUCTIONS:
Directly answer the question with one of the choices: airliner, articulated bus, back, blue, brown, chopper motorcycle, cruiser, cyan, dirtbike, double bus, down, false, fighter aircraft, front, gray, green, jet, left, minivan, mountain bike, purple, red, right, school bus, scooter, sedan, suv, tandem bike, truck, true, up, utility bike, wagon, yellow.

**User:**
Watch the eight frames in the video and answer the question: Is the red mountain moving fast at the beginning?

**GPT-4o:**
false

Figure 10: **Zero-shot evaluation of GPT-4v with multiple choice questions.** We evaluate the proprietary model GPT-4o by accessing the model through OpenAI API with customized system prompts.

---

**Zero-shot evaluation of GPT-4o with step-by-step reasoning**

**System:**
You are an intelligent chatbot designed for answering questions based on eight frames obtained from a video.

Your task is to analyze the frames of the video, identify the nature of the object movements (such as speed, acceleration, and direction), and then determine the answer to the question.

INSTRUCTIONS:
- Think step by step, reason about the dynamic and physical events, and then give the answer.
- End your response with the exact answer to the question, from one of the choices: airliner, articulated bus, back, blue, brown, chopper motorcycle, cruiser, cyan, dirtbike, double bus, down, false, fighter aircraft, front, gray, green, jet, left, minivan, mountain bike, purple, red, right, school bus, scooter, sedan, suv, tandem bike, truck, true, up, utility bike, wagon, yellow.

**User:**
Watch the eight frames in the video. Think step by step and then answer the question: Is the red mountain moving fast at the beginning?

**GPT-4o:**
To determine if the red mountain bike is moving fast at the beginning of the sequence, we need to closely analyze its position in the first few frames:

1. In the first frame, observe the position of the red mountain bike.
2. In the second frame, note any change in position of the red mountain bike.
3. Compare the distance covered by the red mountain bike between these frames.

If the red mountain bike has moved a significant distance between the first two frames, it is moving fast at the beginning. If it has moved only a small distance, it is not moving fast.

From the frames:

- The red mountain bike does not show significant movement between the first two frames.
- The red mountain bike appears to be in almost the same position.
Thus, the red mountain bike does not move fast at the beginning.

The answer is: false.

---

Figure 11: **Zero-shot evaluation of GPT-4v with step-by-step reasoning.** We evaluate the proprietary model GPT-4V by accessing the model through OpenAI API with customized system prompts. Specifically, we encourage the GPT-4o model to think step by step and reason about the dynamic and physical events before predicting the answer.