# OpenReview forum: "Compositional 4D Dynamic Scenes Understanding with Physics Priors for Video Question Answering"
_ICLR.cc/2025/Conference — ICLR 2025 Poster_

### Official Review · Reviewer_kjMd · 2024-10-21

**Soundness:** 3
**Presentation:** 2
**Contribution:** 3
**Rating:** 6
**Confidence:** 3

**Summary:**

This work focuses on the understanding of the dynamic properties of 3D objects in videos. It uses a simulator to control physical concepts (velocity, acceleration, collisions, etc.)  to generate videos, and uses pre-defined programs to generate annotations. Based on the proposed dataset, this work introduces a physics prior based VQA model. The experiment shows its effectiveness on the proposed dataset.

**Strengths:**

1. Overall, the paper is easy-to-follow.
2. The motivation of the dataset is clear.
3. Experiment verifies the effectiveness on the proposed dataset.

**Weaknesses:**

1. The content of the penultimate paragraph is confusing. Firstly, the motivation of the model is unclear. Why an explicit scene representation should be introduced to answer 4D dynamics questions? Secondly, what exactly is the"explicit scene representation"? Thirdly, what is the relation between the "scene parsing module" and "3D generative model", and how do the "3D generative representation" and "symbolic reasoning module" work together?
2. The significance of the proposed dataset is doubtful. In the reviewer's opinion, the scenarios in the proposed dataset may be too limited and fall far short of covering all real-world situations.
3. The effectiveness of the proposed model is also doubtful. The modules are specifically designed based on the actually unknown priors, i.e., the dataset generation and annotation processes. The authors should conduct experiments on spatiotemporal questions of other datasets, including both synthetic and real datasets.

**Questions:**

See weaknesses.

---

> ### Author Response · Authors · 2024-11-22
>
> ### 1. **Motivation and Definition of Explicit Scene Representation**
>   We apologize for any miscommunication and will revise the paper to better define terms like "explicit scene representation," "symbolic reasoning model," and "generative model." We thank the reviewer for their constructive feedback and will incorporate these clarifications in the revised manuscript.
>
>   - **Motivation for Explicit Scene Representation**. As cited in related works like NS-VQA [1], NS-DR [2], and P-NSVQA[3], VR-DP[4], PO3D-VQA[5],  the compositional reasoning model with explicit scene representations has a long history and is proved effective for tasks requiring reasoning about object’s properties and interactions. These previous works demonstrate that the compositional reasoning models are beneficial from the explicit 2D / 3D scene representation in terms of accuracy, interpretability, and robustness for out-of-domain reasoning[3]. This motivation for explicit scene representations is consistent with our aim for 3D dynamical scenes.
>
>     Especially, for our video questions answering tasks, capturing explicit 3D and temporal (4D) dynamics allows us to understand and reason about object interactions in a structured and interpretable way, which is crucial for answering 4D dynamics questions.
>
>   - **What is Explicit Scene Representation**: As written in Section 4.1 (line 308) in the paper, the explicit scene representation is a 3D representation of objects and their properties (e.g., position, velocity, acceleration) over time. This structured representation serves as the foundation for symbolic reasoning.
>
>   - **The concept between "scene parsing module" and "3D generative model"**. As written in Section 4.2 (line 318 to line 323), the scene parsing module uses the 3D generative model as its core component to infer the dynamic properties of the scene. The generative model provides the underlying framework for estimating 3D world states from video frames, while the scene parsing module applies this model to sequential data, integrating temporal and physical reasoning.
>
>  - **How do the "3D generative representation" and "symbolic reasoning module" work together.** The implementation follows the prevailing neural symbolic reasoning pipeline (NS-VQA [1], NS-DR [2], and P-NSVQA[3]). The **3D generative representation** provides a structured, interpretable representation of the scene, including object positions, poses, and physical properties over time. The **symbolic reasoning** module operates on this representation to answer high-level questions step by step. For each step, the operation is implemented by a function, to query the related properties and get the intermediate result, to pass to the next operation [1].
>
> ### **2. Significance of the Proposed Dataset**
>   We appreciate the feedback and agree that the dataset does not cover all real-world scenarios. However, similar to previous benchmarks like CLEVR, CLEVRER and SuperCLEVR, our dataset is intended as a starting point for studying 4D dynamics in a controlled setting. While the scenarios are simplified, they provide a strong foundation for understanding and benchmarking dynamic reasoning tasks.
>
>   Compared to previous datasets (CLEVR, CLEVRER, SuperCLEVR), our dataset offers significantly more realistic 3D assets (agreed by **Reviewer jmWM** ), dynamic properties, and annotated ground truth. Moreover, our framework is designed to be extensible, and future iterations will incorporate more complex and realistic scenarios. We will emphasize these points in the revised manuscript and discuss the extendability of our approach.
>
>   **Reference**
>
> [1] Yi, Kexin, Jiajun Wu, Chuang Gan, Antonio Torralba, Pushmeet Kohli, and Josh Tenenbaum. "Neural-symbolic vqa: Disentangling reasoning from vision and language understanding." Advances in neural information processing systems 31 (2018).
>
> [2] Yi, Kexin, Chuang Gan, Yunzhu Li, Pushmeet Kohli, Jiajun Wu, Antonio Torralba, and Joshua B. Tenenbaum. "Clevrer: Collision events for video representation and reasoning." arXiv preprint arXiv:1910.01442 (2019).
>
> [3] Zhuowan Li, Xingrui Wang, Elias Stengel-Eskin, Adam Kortylewski, Wufei Ma, Benjamin Van Durme, and Alan L Yuille. Super-clevr: A virtual benchmark to diagnose domain robustness in visual reasoning. In Proceedings of the IEEE/CVF Conference on Computer Vision and Pattern Recognition, pp. 14963–14973, 2023.
>
> [4] Ding, Mingyu, Zhenfang Chen, Tao Du, Ping Luo, Josh Tenenbaum, and Chuang Gan. "Dynamic visual reasoning by learning differentiable physics models from video and language." Advances In Neural Information Processing Systems 34 (2021): 887-899.

---

> > ### Author Response · Authors · 2024-11-22
> > **Continuation of the Response**
> >
> > ### **3. The effectiveness of the proposed model**
> > - **Use of Physics Priors**: The physical priors used in the model reflect common-sense world knowledge (e.g., objects follow trajectories governed by physical laws) and are not specific to our dataset alone. These priors provide a generalizable framework for understanding 4D dynamics.
> > - **Cross-Dataset Evaluation**:  Please refer to the section titled “**The Generalization Ability of NS-4DPhysics to Real-World Videos and Question Answering**” in the common reply above.

---

> > > ### Comment · Reviewer_kjMd · 2024-11-25
> > >
> > > Thanks for the responses. Some of my concerns are addressed. However, the effectiveness of the proposed model should be verified with more experiments, and I will keep my original rating.

---

> ### Author Response · Authors · 2024-11-27
> **Addressing Concerns Regarding the Proposed Model**
>
> Regarding the last concern raised in your review, we hope this can be leveraged with real video inference cases as shown in the supplementary material. Due to the constraint of annotation as explained in the previous discussion, we retrain our 3D scene parser on the Pascal3D+ dataset, which contains sufficient 3D annotations but lacks object appearance labels. The detailed results are provided in Fig 1(a) to reconstruct the 3D dynamical world states. We aim to demonstrate that our model can be trained on and infer from real-world data with proper annotations, particularly for the focused 4D dynamical properties. This is feasible if the 3D world states of objects can be successfully estimated. In Fig. (b), we show similar questions and answers as in DynSuperCLEVR in this real-world case. These questions can also be resolved with similar program executions if all properties have been estimated with the proper trained model (as shown in Fig. (c)).
>
> I hope the review acknowledges that real-world video datasets are currently limited by the absence of necessary 3D spatial annotations. This why we start to propose the first benchmark to address this gap in VideoQA, with a focus on 4D dynamics, including velocity, acceleration, and collisions.
>
> It would also be helpful if the reviewer could provide more detailed suggestions about which benchmark you recommend verifying with additional experiments.

---

> > ### Comment · Reviewer_kjMd · 2024-12-03
> >
> > Thanks for the additional qualitative experiments. A solid benchmark model can benefit the community a lot, and that's why I was concerned mostly before. I hope some statistically significant results will be added in the revision. I will raise my rating.

---

### Official Review · Reviewer_gPQm · 2024-11-03

**Soundness:** 2
**Presentation:** 3
**Contribution:** 3
**Rating:** 6
**Confidence:** 3

**Summary:**

The paper addresses the dynamic properties of 3D objects in videos in the task of video question answering. It first proposes a new dataset called DynSuperCLEVR that composes multiple transportation objects into a scene and generates videos of these objects moving. The considered properties are speed, acceleration, and collision. Three types of questions are designed to test VLM's ability to understand the 3D dynamics of objects in these videos. A neural symbolic method, NS-4DPhysics, is proposed to address the importance of explicit 4D representation.

**Strengths:**

- Innovative Dataset: DynSuperCLEVR fills a gap in VideoQA with a focus on 4D dynamics, including velocity, acceleration, and collision, enhancing video-based physics reasoning. The scene is programmed in a way that the ground truth information of object speeds and collision events can be documented and transformed into question-answer pairs.
- Effective Model Design: NS-4DPhysics uses a physics-informed 4D scene representation and neural-symbolic reasoning, excelling in complex VideoQA tasks over baseline models.
- Comprehensive experiments demonstrate NS-4DPhysics’s superior performance in factual, predictive, and counterfactual reasoning.

**Weaknesses:**

- The proposed dataset only spans a narrow domain of scenes and objects and may not generalize well to open-domain scenarios. The CLEVR-like setting makes things look nice and clean; for example, the objects have uniform colors, noise-free textures, rigid objects, and much fewer high-frequency details compared to realistic videos. Method comparison on the dataset may not reflect the true ability of the method in real-world videos.
- The reliance on physics priors might reduce flexibility in scenarios where these priors don’t apply as expected.

**Questions:**

- How did you input the video frames to GPT-4o?

---

> ### Author Response · Authors · 2024-11-23
> **Thank the reviewer for the constructive feedback**
>
> 1. Narrow Domain and Synthetic Nature of the Dataset
>
>     Please refer to a detailed discussion about the synthetic nature of DynSuperCLEVR is in the above common official comment.
>
> 2. Reliance on Physics Priors
>
>     - The physical priors used in the model reflect common-sense world knowledge (e.g., objects follow trajectories governed by physical laws) and are not specific to our dataset alone. Also, as  shown in the paper, we conducted ablation studies demonstrating that our model performs well even without physics priors, albeit with some degradation in performance. This highlights the model's robustness and flexibility in scenarios where the priors may not fully apply.
>
>     - We are working on the more abolition study, which compare the performance under different physical parameters, where the results show our results is robust to the strength physical prior choice.
>
>
>
> 3. How did you input the video frames to GPT-4o?
>
>     We downsampled the video to 8 frames for computational efficiency. These frames were then processed and sent to the GPT-4o API according to the guidelines outlined in the OpenAI GPT-4o documentation (https://cookbook.openai.com/examples/gpt4o/introduction_to_gpt4o).

---

### Official Review · Reviewer_oBU1 · 2024-11-04

**Soundness:** 3
**Presentation:** 3
**Contribution:** 3
**Rating:** 6
**Confidence:** 4

**Summary:**

This paper makes two main contributions: (1) introducing DynSuperCLEVR, a novel video question answering dataset that focuses on understanding 4D dynamics (velocity, acceleration, collisions) of objects in 3D scenes, and (2) proposing NS-4DPhysics, a neural-symbolic model that integrates physics priors with 3D scene understanding for dynamics reasoning. Through extensive experiments, their model significantly outperforms existing approaches, including large multimodal models, demonstrating current limitations in physical reasoning capabilities of video-language models.

**Strengths:**

- The paper's main objective of addressing multimodal 4D dynamics understanding is well-motivated.
- The authors provide comprehensive evaluation results across three types of reasoning tasks (factual, predictive, and counterfactual), demonstrating the model's capabilities in different scenarios.
- The proposed physics-aware neural-symbolic architecture presents an innovative approach

**Weaknesses:**

- The dataset only considers rigid objects with linear velocity and acceleration. Real-world scenarios often involve more complex dynamics like non-rigid deformation, rotation-based motion, and fluid dynamics.
- The dataset uses synthetic rendering which may not capture real-world challenges like motion blur, camera shake, varying lighting conditions, and partial occlusions.
- The ablation studies are limited. While the paper shows the importance of physics priors, there could be more detailed analysis of other architectural choices and hyperparameters, like the impact of different CNN backbones or the choice of physics engine parameters.

**Questions:**

- It would be great if the author analyze the performance of proposed model on more complex dynamic scenarios such as non-rigid object deformation or fluid dynamics
- How do the authors plan to address the challenges in real datasets such as motion blur, camera shake, and varying lighting conditions?
- How do architectural choices  or important hyperparameters such as different CNN backbones or the choice of physics engine parameters impact on the performance of proposed model?

---

> ### Author Response · Authors · 2024-11-23
> **We appreciate the reviewer’s insightful feedback**
>
> 1. **Limited Dynamics (Rigid Objects and Linear Motion)**
>   Please refer to the common question in the Official Comments above for the discussion on object diversity.
>
> 2. **Synthetic Rendering and Real-World Challenges**
>
>     The challenges of synthetic data generalization, such as motion blur, camera shake, and lighting variability, are indeed important. Similar to widely used datasets like CLEVR and CLEVRER, our approach focuses on controlled synthetic benchmarks to establish a strong foundation for understanding 4D dynamics in vision-language models.
>
>     To address these concerns, we have created a **subset of the dataset** that incorporates varying lighting conditions and motion blur. The original DynSuperCLEVR dataset already utilizes realistic ambient lighting from HDRI environment maps, including both daytime and nighttime scenes. During the rebuttal phase, we introduced additional variations by controlling the strength of ambient light to increase variability.
>
>     To mimic motion blur, for each frame $ F_i $, we created a motion-blurred frame $ F_i' $ by averaging pixel values across a temporal window of random size (0–5) centered on $ F_i $. This subset serves as a bridge between synthetic and real-world scenarios, enabling the evaluation and improvement of model robustness in more challenging conditions.
>
> 3. **Limited Ablation Studies**
>
>     We appreciate the reviewer’s suggestion to analyze the impact of additional architectural choices and hyperparameters. While our current ablation studies focus on the importance of physics priors, we have included new experiments to evaluate the effects of different CNN backbones and variations in the physics engine parameter $\sigma$ (see the Equation (2) in main paper). Our preliminary findings suggest that the model maintains robustness across these changes, with the best performance observed at $\sigma=2$, as used in the submission. Performance across different CNN backbones is comparable (all above 77, with ResNet50 slightly lower). A summary of the results is provided below:
>
> | Physics Prior $\sigma$      | AVG   | All   | Vel.  | Acc.  | Col.  | Predictive | Counterfactual |
> |-----------------------------|-------|-------|-------|-------|-------|------------|----------------|
> | 4                           | 80.46 | 85.24 | 87.06 | 82.01 | 84.03 | 83.73      | 72.41          |
> | 2                           | 82.64 | 87.70 | 88.66 | 83.73 | 88.46 | 85.71      | 74.51          |
> | 1                           | 78.01 | 82.19 | 84.26 | 81.71 | 78.31 | 81.30      | 70.53          |
> | 0.5                         | 74.91 | 79.79 | 81.89 | 80.64 | 74.82 | 74.88      | 70.07          |
> | w/o physics prior   | 75.97 | 79.68 | 81.40 | 81.30 | 74.88 | 78.83      | 69.39          |
>
> | Model Backbone              | AVG   | All   | Vel.  | Acc.  | Col.  | Predictive | Counterfactual |
> |-----------------------------|-------|-------|-------|-------|-------|------------|----------------|
> | Resnet50                   | 77.60 | 82.20 | 83.57 | 76.22 | 84.11 | 80.57      | 70.04          |
> | ViT8                        | 81.90 | 87.34 | 88.26 | 83.54 | 88.46 | 84.79      | 73.58          |
> | Resnext2                    | 82.64 | 87.70 | 88.66 | 83.73 | 88.46 | 85.71      | 74.51          |
> ---
>
> ### Questions
>
> 1. **Performance on Complex Dynamics (e.g., Non-Rigid Deformation, Fluid Dynamics)**
>
>     Please refer to the Official Comments for the discussion about non-rigid objects in the dataset.
>
>     Extending the model to handle non-rigid deformations and fluid dynamics is a meaningful direction for future work. While this paper focuses on rigid body dynamics,  the core component 3D generateive model has been demonstrated the possibility in applying on deformable setting [1] and we are actively exploring these extensions.
>
>     [1] Wang, Angtian, Wufei Ma, Alan Yuille, and Adam Kortylewski. "Neural textured deformable meshes for robust analysis-by-synthesis." In Proceedings of the IEEE/CVF Winter Conference on Applications of Computer Vision, pp. 3108-3117. 2024.
>
> 2. **Addressing Real-World Challenges (e.g., Motion Blur, Camera Shake, Lighting Variability)**
>
>      As discussed above, we explored extensions to the dataset with realistic variations, including motion blur, varying lighting conditions to improve the model's robustness in real-world scenarios.
>
> 3. **Impact of Architectural Choices and Hyperparameters**
>
>     Please see the discussion of weakness 1 above for these ablation studies.

---

> > ### Comment · Reviewer_oBU1 · 2024-11-26
> >
> > Thank you to the author for the clarification. I will maintain my original rating and lean to accept this paper.

---

### Official Review · Reviewer_jmWM · 2024-11-04

**Soundness:** 3
**Presentation:** 3
**Contribution:** 3
**Rating:** 6
**Confidence:** 2

**Summary:**

The paper introduces DynSuperCLEVR, a video question answering (VideoQA) dataset that emphasizes understanding dynamic 3D object properties within 4D (3D + time) scenes.
Additionally, the authors present NS-4DPhysics, a model that combines neural-symbolic reasoning with physics-based priors to analyze these dynamic properties. The model first constructs an explicit 4D scene representation using a 3D generative model, followed by neural-symbolic reasoning to answer questions.
Experimental results demonstrate that NS-4DPhysics surpasses existing VideoQA models across various question types (factual, predictive, and counterfactual), underscoring its effectiveness in reasoning about object dynamics in complex, synthetic environments.

**Strengths:**

**1. Novel Dataset:** DynSuperCLEVR is a novel dataset that focuses on 4D dynamics, addressing a critical gap in existing VideoQA datasets which typically overlook explicit physics-based scene understanding.

**2. Innovative Model Design:** The NS-4DPhysics model combines 3D generative modeling with physics-informed priors, represents an innovative approach to handling dynamic 4D scene reasoning.

**3. Comprehensive Benchmarking:** Extensive evaluations against baseline models, including video large language models (Video-LLMs) and other symbolic frameworks, highlight the superior performance of NS-4DPhysics in capturing 4D dynamics.

**4. Future and Counterfactual Simulations:** By leveraging physics-based priors, the model excels at simulating both future and hypothetical states, demonstrating practical value and broad application potential.

**Weaknesses:**

**1. Synthetic Data Limitations:** While the dataset is suitable for testing dynamic properties, its synthetic nature may limit generalizability to real-world applications. Despite the authors’ efforts to improve aspects like background realism (L201), models trained exclusively on synthetic data often struggle to handle real-world noise and variability.

**2. Computational Complexity:** The NS-4DPhysics model is computationally demanding due to its reliance on 3D generative modeling and physics-based priors, presenting challenges for scalability and use in resource-constrained environments.

**3. Limited Object Diversity:** The dataset is limited to a narrow range of rigid objects, which may not adequately represent the complexity of real-world scenes that often include deformable or articulated objects.

**4. Evaluation of Real-World Applicability:** The paper lacks an analysis of the model’s performance on real-world video data, which is essential for evaluating its practical applicability outside synthetic benchmarks.

**Questions:**

1. Have the authors considered extending the dataset to include articulated or deformable objects? If so, what challenges or limitations do they anticipate with this extension?

2. Could the authors provide an efficiency analysis of the proposed model, including resource usage and runtime under typical conditions?

3. What modifications to the NS-4DPhysics framework would make it more efficient for real-time performance or deployment in resource-limited environments?

---

> ### Author Response · Authors · 2024-11-22
> **Thanks for the feedback!**
>
> We appreciate the reviewer’s insightful feedback and will incorporate these clarifications and additions in the revised manuscript.
>
>
> 1. **Synthetic Data Limitations**
>
>     A more detailed discussion about the synthetic nature of DynSuperCLEVR is in the above common official comment.
>
>     We acknowledge the importance of generalizing to real-world scenarios and appreciate the reviewer recognizing our efforts to enhance the realism of textures in the synthetic dataset. Regarding realism, we believe it is entirely plausible to make this dataset increasingly realistic in future developments.
>
> 2. **Computational Complexity.**
>
>     We add a detailed efficiency analysis of the average cost of NS-4DPhysics model in a video clip of 120 frames, with 2 second,  with 5 objects in the scenes.
>
>     **Time Efficiency:** For estimating dynamical properties for each frame, the total time for the 3D scene parser is 557.85 ms, with 3D generative modeling taking 521.15 ms and physics prior calculation requiring only 21.49 ms. For each video clip, predicting static properties across the entire video takes 140.16 ms, while the question parser and program execution take 5.15 ms and 0.03 ms, respectively.
>
>     **Computation Usage:** As the bottleneck of the pipeline, the peak virtual memory occupation for 3D generative modeling, dynamical properties estimation and physics prior calculation is 3179 MiB.
>
> 3. **Limited Object Diversity**.
>
>     We agree that extending the dataset to include articulated or deformable objects is a meaningful direction for real-world simulation. However, our primary focus is on rigid body dynamics as a foundational step toward investigating fundamental dynamic properties in vision-language tasks. Even without deformable objects, we have made several important findings: 1) the dataset effectively reveals the limitations of existing models in understanding fundamental dynamics, and 2) explicitly representing 4D dynamic properties, as implemented in the proposed NS-4DPhysics model, significantly enhances model performance.
>
>     Incorporating articulated or deformable objects introduces significant challenges, such as complex annotations and increased computational costs. We plan to address these challenges in future work and will note this in the paper to clarify our scope and future directions.
>
>
> 4. **Evaluation of Real-World Applicability**.
>
>    Please refer to the third point in the common questions section of the official comments above, which discusses the generalization ability of our proposed model, NS-4DPhysics
>
> 5. **Have the authors considered extending the dataset to include articulated or deformable objects?**
>
>     Please refer to the previous discussion in "**3. Limited Object Diversity**" above.
>
> 6.  **Could the authors provide an efficiency analysis of the proposed model, including resource usage and runtime under typical conditions?**
>
>     Please refer to the previous discussion in "**2. Computational Complexity**"
>
> 7. **What modifications to the NS-4DPhysics framework would make it more efficient for real-time performance or deployment in resource-limited environments?**
>
>      As discussed above, the current model processes approximately 2 frames per second, which corresponds to handling a 2 fps video stream in real-time. To improve real-time performance at higher frame rates, we can utilize lightweight 3D representations within the neural mesh modeling framework for faster inference. Furthermore, while the current DynSuperCLEVR is setup at a frame rate of 60, the model is also applicable for lower frame rate scenarios, as the dynamic properties of the video are processed frame-wise. Additionally, our proposed physical prior shows superior performance in preserving temporal consistency, as demonstrated in Sections 5.2 and 5.3.

---

> > ### Comment · Reviewer_jmWM · 2024-11-26
> >
> > Thank you for your thorough rebuttal. Your responses have effectively addressed my concerns, and I am inclined to accept the paper.

---

### Author Response · Authors · 2024-11-22
**We sincerely thank all reviewers for their time and efforts in reviewing the paper.**

We are happy that the reviewers recognized the following key strengths and contributions of our work:

1. **Novel Dataset and Task**. The introduction of **DynSuperCLEVR** as a novel dataset addresses a critical gap in VideoQA research by focusing on 4D dynamics, including explicit physics-based scene understanding ("a crucial aspect of visual reasoning" (jmWM),  "fills a gap in VideoQA with a focus on 4D dynamics" (gPQm), "well-motivated" (oBU1, kjMd)).

2. **Innovative Model Design**. The **NS-4DPhysics** model combines 3D generative modeling with physics-informed priors, offering an effective approach to dynamic 4D scene reasoning ("represents an innovative approach" (jmWM), "physics-aware neural-symbolic architecture" (oBU1), "excels in complex VideoQA tasks" (gPQm)).

3. **Comprehensive Evaluations**. Experiments across factual, predictive, and counterfactual reasoning tasks demonstrate NS-4DPhysics’s superior performance in understanding 4D dynamics and outperforming baseline models ("highlight the superior performance of NS-4DPhysics" (jmWM), "comprehensive evaluation results" (oBU1), "effective in reasoning tasks over baselines" (gPQm)).

In the following, we discuss common questions among the reviewers:

1. **The synthetic nature of the DynSuperCLEVR dataset and the domain gap between real-world scenes. (Reviewer  jmWM, oBU1, gPQm, kjMd)**

    We acknowledge the importance of generalizing to real-world scenarios and appreciate the reviewers recognizing our efforts to enhance the realism of textures in the synthetic dataset (jmWM). While real-world annotations are extremely challenging to obtain, our primary goal, similar to popular datasets like CLEVRER and COMPHY, is to provide a controllable environment with fully annotated 3D dynamic properties. This dataset serves as a fundamental benchmark for studying how vision-language models understand these properties in simplified scenarios, offering insights that are difficult to achieve with real-world data. We believe it is plausible to make this dataset increasingly realistic in future iterations, but this effort would go beyond the scope of this work.

2. **Extension to non-rigid objects. (Reviewer jmWM, oBU1, gPQm)**

    We agree that extending the dataset to include articulated or deformable objects is an important direction for real-world simulation. However, our current focus is on rigid body dynamics as a foundational step for understanding fundamental dynamic properties in vision-language tasks. Even without deformable objects, we have made significant findings: 1) the dataset effectively highlights the limitations of existing models in understanding fundamental dynamics, and 2) explicitly representing 4D dynamic properties, as implemented in NS-4DPhysics, greatly enhances model performance. While incorporating deformable objects poses would likely enable us to study even more advanced questions, we plan to address these in future work due to challenges like complex annotations and higher computational costs.

 3. **The generalization ability of NS-4DPhysics to real-world videos and question answering. (Reviewer  jmWM, oBU1, gPQm,kjMd)**

    The primary aim of NS-4DPhysics is to extend compositional reasoning models (e.g., NS-VQA, NS-VR, PO3D-VQA) to dynamic property reasoning tasks by leveraging explicit 4D scene representations. While its application to real-world video datasets is currently limited by the absence of necessary 3D spatial annotations, the 3D generative model used in our scene parser demonstrates strong 6D pose estimation capabilities on real-world data [1] and promising out-of-domain adaptation ability [2]. Since NS-4DPhysics builds upon this foundational work, we expect it to generalize effectively to real-world scenarios.

**References**

[1] Ma, Wufei, Angtian Wang, Alan Yuille, and Adam Kortylewski. "Robust category-level 6d pose estimation with coarse-to-fine rendering of neural features." In European Conference on Computer Vision, pp. 492-508. Cham: Springer Nature Switzerland, 2022.

[2] Kaushik, Prakhar, Aayush Mishra, Adam Kortylewski, and Alan L. Yuille. 2024. "Source-Free and Image-Only Unsupervised Domain Adaptation for Category-Level Object Pose Estimation." International Conference on Learning Representations (ICLR) 2024.

---

> ### Author Response · Authors · 2024-11-27
> **To support the generalization ability to real-world scenarios**
>
> Here we aim to provide additional evidence demonstrating the generalization ability of our proposed model, NS-4DPhysics, to real-world videos. We consider a case study experiment, as detailed in the new supplementary material.
>
> We consider a real-world video with the collision event.  We retrain the 3D scene parser on the Pascal3D+ dataset, which includes 3D pose annotations but without object appearance labels.
> The qualitative reconstruction results are provided in the supplementary material. As shown in Fig.1(a), we visualize the accurate estimations of cars from 3D scenes on the real video, where the 4D dynamic properties, including velocities, accelerations and collisions, can be inferred explicitly. Although the model is not trained on object appearances, its capabilities can be further enhanced by incorporating object classifiers with proper annotations or enabling open-vocabulary recognition through pre-trained large vision-language feature embeddings (e.g., CLIP). This represents an important direction for future work. Additionally, as shown in Fig.1(b) and (c), similar types of questions can be posed for the given video, which can then be answered by executing the program step-by-step.

---

### Meta-Review · Area_Chair_tPfc · 2024-12-22

**Metareview:**

This work proposes a new VideoQA dataset that emphasizes understanding dynamic 3D object properties within 4D. All reviewers consistently recommended accepting this work. AC agrees that this work is interesting and deserves to be published on ICLR 2O25. The reviewers did raise some valuable concerns that should be addressed in the final camera-ready version of the paper. The authors are encouraged to make the necessary changes in the final version.

**Additional Comments On Reviewer Discussion:**

All reviewers consistently recommended accepting this work.

---

### Decision · Program_Chairs · 2025-01-22

Accept (Poster)